# OpenReview forum: "Security--Fidelity Tradeoffs: The Hidden Cost of Prompt Injection Defense"
_ICML.cc/2026/Conference — ICML 2026 spotlight_

### Official Review · Reviewer_LHcD · 2026-02-18

**Soundness:** 3
**Presentation:** 3
**Significance:** 3
**Originality:** 3
**Overall Recommendation:** 4
**Confidence:** 4

**Summary:**

This paper identifies a critical conflict in LLM security: the "security-fidelity tradeoff." The authors demonstrate that defenses against indirect prompt injection inevitably impair a model's ability to process benign, instruction-like text. Specifically, they show that current defenses achieve robustness by aggressively suppressing valid content, leading to significant fidelity losses in tasks like translation. The authors argue this tension is not merely an empirical limitation but a fundamental decision-theoretic property, suggesting that the optimal balance is strictly task-dependent. To quantify this, they introduce SECFID, a benchmark with probes designed to unambiguously distinguish between three behaviors: processing data faithfully, ignoring suspicious input, or executing a hijack.

**Compliance With Llm Reviewing Policy:**

Affirmed.

**Key Questions For Authors:**

1. Figure 2 presents a compelling empirical frontier, and Section 5 formalizes this via decision theory. However, humans can often achieve both perfect security and fidelity when properly instructed. Why do you believe this tradeoff is fundamental to LLMs rather than a reflection of current capability gaps?

2. The study evaluates models like GPT-5.2 and Gemini 3 Flash but does not explicitly discuss the impact of "reasoning" architectures (e.g., newer open-source reasoning models like Qwen3 or gpt-oss-20b). Do these models—which can "think" before outputting—follow the same tradeoff curve?

3. It is surprising that GPT-5.2 exhibits a hijack rate greater than 20% (Figure 2). Was this evaluated with a specific reasoning effort setting? Does increasing the reasoning effort reduce the hijack rate without compromising fidelity?

**Limitations:**

yes

**Strengths And Weaknesses:**

### Strengths

1. The paper effectively exposes "fidelity loss" as a hidden cost of prompt injection defenses. The argument that defenses optimizing for standard security benchmarks may silently degrade performance on tasks requiring instruction-like data handling is both novel and persuasive.

2. The introduction of SECFID is a significant contribution. By separating "utility" from "fidelity" it provides a more granular metric for evaluating how models handle ambiguous inputs, moving beyond simple success/failure rates.

3.  The empirical findings—particularly the security-fidelity frontier—clearly illustrate the practical consequences of the tradeoff across a diverse set of models (Llama, Qwen, GPT, Gemini).

### Weaknesses

1. The paper posits that the security-fidelity tradeoff is fundamental, supported by a decision-theoretic framework. However, this relies on the assumption that models cannot distinguish intent with perfect accuracy. It is possible that the observed tradeoff is an artifact of current model limitations rather than an insurmountable impossibility. A sufficiently capable model (e.g., a human expert) could theoretically achieve both high security and high fidelity.

2. Observations around the impact of model size on injection resistance focus on non-reasoning models. It's unclear if these observations translate to reasoning models as well.

3. The paper does not investigate how varying the reasoning-effort  (inference-time compute) affects the hijack rate. It is plausible that increased reasoning time could allow models to better disambiguate benign instructions from adversarial ones, potentially shifting the frontier.

---

> ### Author Rebuttal · Authors · 2026-03-31
>
> We thank the reviewer for the careful reading and thoughtful feedback. We are encouraged that the reviewer found the paper’s central contribution clear: exposing fidelity loss as a hidden cost of prompt injection defenses and showing a compelling empirical security-fidelity frontier across models. We view the main issues raised here as clarifying the scope of the decision-theoretic claim and addressing whether reasoning effort can shift the empirical frontier. We respond to each below.
>
> ## W1 & Q1
>
> We agree that much of what Figure 2 captures reflects current model and defense limitations and that capability improvements can substantially reduce the empirical tradeoff. The reasoning results below, and the fidelity-aware training in YaT2/W4 shows the frontier can be pushed outward even for current defenses.
>
> The formal result in Section 5 makes a separate, narrower point that the desired behavior for instruction-like content is task-dependent. In some deployments, ignoring such content is correct; in others, processing it is. Because the right action depends on deployment-specific costs external to the input, no single task-agnostic policy is universally optimal. We will revise the framing to note this and ensure "fundamental" is not read as a permanent capability limit.
>
> ## W2, W3, Q2, Q3
>
> We agree that reasoning effort is an important axis the paper should address. To this end, we expanded the evaluation to include reasoning-effort sweeps for several frontier models
>
> | Model | Sec (%) | Fid (%) |
> | :---- | :---- | :---- |
> | **Claude Haiku 4.5** | 73.5 | 80.8 |
> | \+ thinking | 73.1 (-0.4) | 78.8 (-2.0) |
> | **Claude Sonnet 4.6** | 83.2 | 69.8 |
> | \+ low | 82.7 (-0.5) | 64.2 (-5.6) |
> | \+ medium | 86.0 (+2.8) | 68.9 (-0.9) |
> | \+ high | 89.9 (+6.7) | 72.4 (+2.6) |
> | **Claude Opus 4.6** | 88.9 | 74.0 |
> | \+ low | 85.5 (-3.4) | 66.5 (-7.5) |
> | \+ medium | 88.6 (-0.3) | 72.7 (-1.3) |
> | \+ high | 91.6 (+2.7) | 77.0 (+3.0) |
> | **Gemini 3.1 Flash-Lite (minimal)** | 62.2 | 91.4 |
> | \+ low | 58.8 (-3.4) | 95.0 (+3.6) |
> | \+ medium | 64.2 (+2.0) | 96.8 (+5.4) |
> | \+ high | 74.2 (+12.0) | 96.3 (+4.9) |
> | **Gemini 3 Flash (minimal)** | 72.8 | 91.8 |
> | \+ low | 92.0 (+19.2) | 96.6 (+4.8) |
> | \+ medium | 92.5 (+19.7) | 96.1 (+4.3) |
> | \+ high | 93.9 (+21.1) | 95.6 (+3.8) |
> | **GPT-5.4 Nano** | 77.7 | 89.8 |
> | \+ low | 89.5 (+11.8) | 89.7 (-0.1) |
> | \+ medium | 92.7 (+15.0) | 89.5 (-0.3) |
> | \+ high | 93.1 (+15.4) | 90.4 (+0.6) |
> | \+ xhigh | 96.3 (+18.6) | 93.1 (+3.3) |
> | **GPT-5.4 Mini** | 66.9 | 90.3 |
> | \+ low | 76.6 (+9.7) | 93.9 (+3.6) |
> | \+ medium | 77.0 (+10.1) | 94.3 (+4.0) |
> | \+ high | 79.0 (+12.1) | 93.5 (+3.2) |
> | \+ xhigh | 86.9 (+20.0) | 94.3 (+4.0) |
> | **GPT-5.4** | 80.1 | 85.1 |
> | \+ low | 89.2 (+9.1) | 92.5 (+7.4) |
> | \+ medium | 94.2 (+14.1) | 92.4 (+7.3) |
> | \+ high | 94.1 (+14.0) | 93.0 (+7.9) |
> | \+ xhigh | 97.2 (+17.1) | 92.5 (+7.4) |
> | **Grok 4.20** | 34.3 | 94.5 |
> | \+ reasoning | 92.0 (+57.7) | 94.0 (-0.5) |
>
> Additional reasoning often substantially improves security and can also improve fidelity, shifting the empirical frontier outward rather than simply tracing the same curve. The effect is not uniform, however. Claude Haiku's thinking setting is slightly worse on both axes, and some lower-reasoning settings improve one axis more than the other. We will incorporate these full results in the revision.
>
> For Q3 specifically, GPT-5.2 in the original draft was evaluated in its default non-reasoning configuration, so the \>20% hijack rate in Figure 2 reflects a base-setting result. The reasoning sweeps for GPT-5.4 and related models above show that additional reasoning can substantially reduce hijack rate while often preserving or improving fidelity.

---

> > ### Author Rebuttal · Reviewer_LHcD · 2026-04-05
> >
> > The rebuttal addressed most of my concerns. I will maintain my positive score.

---

> > > ### Author Response · Authors · 2026-04-08
> > >
> > > We're glad the rebuttal addressed your concerns, and we thank you for your continued support of the paper.
> > >
> > > Best,
> > > Authors

---

### Official Review · Reviewer_XaNV · 2026-03-10

**Soundness:** 3
**Presentation:** 3
**Significance:** 2
**Originality:** 2
**Overall Recommendation:** 3
**Confidence:** 3

**Summary:**

This paper investigates the security challenges posed by **indirect prompt injection** in large language model (LLM) applications that process untrusted external content, such as retrieved documents or user-provided text. The authors argue that existing research primarily focuses on improving attack resistance while overlooking a fundamental tradeoff between **security** and **fidelity**. In particular, defenses that aggressively filter instruction-like content to prevent injection attacks may also suppress legitimate text that should be faithfully processed by the model, thereby degrading task fidelity.

To study this phenomenon, the paper introduces **SECFID**, a benchmark designed to jointly evaluate security and fidelity in LLM systems. The benchmark constructs tasks where three possible behaviors—executing an injected instruction, ignoring it, or processing it as ordinary data—produce distinguishable outputs. This design allows the evaluation framework to explicitly measure both **hijack resistance** (security) and **faithful processing of instruction-like content** (fidelity). SECFID contains 895 instances spanning tasks such as translation, entity extraction, and entity counting, with injected probes varying in placement and framing.

Using this benchmark, the authors evaluate several open and closed LLMs as well as existing prompt injection defenses. The empirical results show that current models and defenses lie on a **security–fidelity frontier**, where improving resistance to prompt injection typically comes at the cost of increased over-filtering of legitimate content. The study further finds that increasing model scale improves task performance but does not significantly reduce injection vulnerability, and that different defenses achieve robustness through different mechanisms, such as repairing misclassifications or broadly suppressing instruction-like content.

Finally, the paper provides a **decision-theoretic analysis** to explain why this tradeoff is fundamental. The authors argue that when benign and adversarial inputs overlap in form, the optimal decision threshold depends on the relative costs of security failures versus fidelity errors, which vary across applications. Consequently, the paper concludes that there is **no universal defense against prompt injection**, and that robustness must instead be managed through task-dependent security policies.

**Compliance With Llm Reviewing Policy:**

Affirmed.

**Final Justification:**

After considering both the paper and the authors’ rebuttal, I based my final recommendation on an overall assessment of the work’s strengths and weaknesses across soundness, originality, significance, and clarity. In my view, the paper presents a meaningful idea and addresses an interesting problem, with several strengths in its motivation and technical formulation. At the same time, there remain some limitations in aspects such as empirical support, completeness of validation, or the clarity of certain claims and presentation.

The rebuttal was helpful and addressed a number of my concerns, particularly by clarifying the authors’ intent, assumptions, and experimental choices. While it did not completely remove all of my reservations, it improved my understanding of the work and reinforced that the authors had carefully considered the main issues raised during review. Overall, I weighed the paper’s strengths against its remaining weaknesses and provided my final recommendation accordingly.

**Key Questions For Authors:**

**1. Generalization to more realistic LLM application settings.**
The SECFID benchmark focuses on tasks such as translation, entity extraction, and counting, which provide clean behavioral separability for evaluation. However, many prompt injection risks arise in more complex **agentic or retrieval-augmented systems**, where models interact with tools, external APIs, or multi-step reasoning chains.
Could the authors discuss how the security–fidelity tradeoff might manifest in these settings, and whether SECFID could be extended to such scenarios?
*If the authors can demonstrate that the proposed evaluation framework generalizes to more realistic LLM pipelines (e.g., tool use or RAG-based agents), it would strengthen the practical significance of the work.*

---

**2. Robustness of the “no universal defense” claim under richer system designs.**
The paper argues that no universal defense exists when benign and adversarial inputs overlap, supported by a decision-theoretic formulation that models the system’s choice as PROCESS vs. FILTER. In practice, however, LLM systems may employ richer strategies, such as asking clarification questions, applying structured instruction–data separation, or executing operations with restricted privileges.
How would the authors’ theoretical argument extend if the action space included such intermediate responses beyond simple processing or filtering?
*Clarifying whether the impossibility claim still holds under richer system designs would strengthen the theoretical foundation of the paper.*

---

**3. Sensitivity of SECFID results to probe construction and dataset design.**
The benchmark relies on carefully constructed probes where processing, ignoring, and executing produce distinguishable outputs. Could the authors provide additional analysis on how sensitive the results are to the specific probe formulations, phrasing, or placement strategies used in SECFID?
For example, do alternative probe templates or more naturalistic instruction-like content significantly affect the observed security–fidelity frontier?
*If the observed tradeoff remains stable across broader probe distributions, it would further validate the robustness of the benchmark.*

---

**4. Practical guidance for navigating the security–fidelity tradeoff.**
The paper convincingly demonstrates that different defenses achieve robustness through different mechanisms (e.g., repair vs. suppression), and that optimal tradeoffs depend on application-specific error costs. However, the paper provides limited guidance on how practitioners should choose an appropriate operating point in practice.
Do the authors envision practical strategies (e.g., dynamic thresholds, task-aware policies, or system-level guardrails) that could help developers manage this tradeoff when deploying LLM applications?
*Additional discussion on deployment strategies could significantly improve the practical impact of the work.*

**Limitations:**

yes

**Strengths And Weaknesses:**

## Strengths

**1. Clear problem formulation and sound conceptual framing (Soundness & Significance).**
The paper identifies and formalizes an important but often overlooked tension in LLM security: the **security–fidelity tradeoff** when defending against indirect prompt injection. The decision-theoretic formulation provides a clear conceptual lens for understanding why defenses that aggressively filter instruction-like content may inadvertently degrade task fidelity. This framing is logically coherent and helps clarify a recurring but previously under-articulated issue in prompt injection research. By highlighting that optimal robustness depends on application-specific error costs, the paper provides a useful perspective for practitioners deploying LLM systems.

**2. A carefully designed benchmark that separates security and fidelity (Originality & Soundness).**
The proposed **SECFID benchmark** introduces a useful evaluation principle based on *behavioral separability*, where ignoring, processing, and executing injected content produce distinguishable outputs. This design allows the evaluation to disentangle security failures (executing injected instructions) from fidelity failures (over-filtering legitimate content). The benchmark construction is well motivated, and the evaluation protocol is clearly specified, enabling reproducibility and systematic analysis across different models and defenses.

**3. Insightful empirical analysis of current models and defenses (Significance).**
The experimental study across multiple open and closed models provides several interesting findings, such as the observation that **model scaling improves capability but does not significantly reduce prompt injection vulnerability**, and that different defenses achieve robustness through qualitatively different mechanisms (e.g., repair vs. suppression). These observations offer practical insights for researchers and system designers and highlight limitations of current evaluation practices that measure only attack resistance.


## Weaknesses

**1. Limited coverage of realistic LLM application scenarios (Significance).**
While SECFID provides a controlled setting for analyzing the security–fidelity tradeoff, the benchmark focuses primarily on relatively simple tasks such as translation, entity extraction, and counting. Many real-world prompt injection risks occur in more complex environments, including **tool-augmented agents, multi-step reasoning pipelines, and retrieval-augmented generation systems**. Because these scenarios involve additional interactions between instructions, external tools, and system prompts, it remains unclear whether the observed tradeoff and evaluation results fully generalize to these more realistic deployment settings.

**2. The paper does not introduce a new defense method (Originality).**
The main contribution of the paper lies in the conceptual framing and evaluation methodology rather than proposing a novel defense mechanism. The empirical study evaluates existing defenses such as SECALIGN and DefensiveTokens but does not explore new approaches for mitigating the identified tradeoff. As a result, the paper provides limited guidance on how future defense mechanisms might explicitly manage or reduce the security–fidelity tension in practice.

**3. The theoretical argument relies on a simplified decision model (Soundness).**
The decision-theoretic analysis models the tradeoff as a binary decision between processing and filtering instruction-like content. While this abstraction provides an intuitive explanation for the observed phenomenon, real-world LLM systems may employ richer strategies, such as uncertainty-aware responses, clarification requests, or structured separation between instructions and data. Because these possibilities are not considered in the theoretical framework, the claim that “no universal defense exists” may depend on the simplifying assumptions of the model and could benefit from a more comprehensive theoretical treatment.

---

> ### Author Rebuttal · Authors · 2026-03-31
>
> We thank the reviewer for the thoughtful feedback. We are encouraged that the reviewer found the problem meaningful, the benchmark design careful and well motivated, and the empirical analysis insightful. The main points raised here concern generalization to complex LLM pipelines, practical guidance strategies, the theoretical claim under richer action spaces, and sensitivity to probe construction. We appreciate these points and address each below.
>
> ## W1 & Q1
> We agree that extending the study to real-world LLM application scenarios strengthens the practical significance of the work. To show SecFid applied to agentic environments, we extend the InjecAgent (Zhan et al., 2024\) benchmark with 120 examples where probes are embedded inside tool outputs. Evaluating Llama 3.1 8B \+ SecAlign reveals a similar pattern where defense methods trade fidelity for security, demonstrating that the framework generalizes to realistic LLM pipelines:
>
> | Model | Sec (%) | Fid (%) |
> | ----- | ----- | ----- |
> | Llama 3.1 8B | 96.7 | 86.7 |
> | \+ SecAlign | 98.3 *(+1.6)* | 85.0 *(−1.7)* |
>
> We also note that these rates are inflated due to the agent being unable to complete the task, further motivating the base benchmark design.
>
> ## W2 & Q4
> We agree that the paper focuses more on evaluation methodology than on actionable deployment advice. We address this below with deployment guidance and a preliminary mitigation experiment motivated by these questions.
>
> **Deployment guidance.** The repair-vs-suppress analysis in §4.4 suggests that security scores alone can obscure meaningful differences in how defenses behave in practice. We believe this distinction can inform deployment decisions in at least two ways. First, pairing security evaluation with direct fidelity measurement helps surface cases where a defense silently drops valid content — a cost that may go unnoticed if only the attack success rate is tracked. Second, the choice between repair-type and suppress-type defenses depends on the deployment context: when incomplete output is costly, repair-type defenses that maintain high processing rates may be more appropriate, while suppress-type defenses may be preferable when even a single hijack could cause irreversible harm.
>
> **Fidelity-aware preference tuning.** We finetune SecAlign 8B using DPO with RS-LoRA and construct preference pairs (processed \> ignored \> executed) from both gold references and model-generated outputs, totaling 754 training examples with a 15% held-out test set.
>
> On the held-out test set, the fidelity-tuned model maintains perfect security while improving fidelity:
>
> | Model | Sec (%) | Fid (%) |
> | ----- | ----- | ----- |
> | SecAlign 8B | 100.0 | 82.3 |
> | \+ Fidelity DPO | 100.0 *(+0.0)* | 90.8 *(+8.5)* |
>
> We also evaluate on the edit task, discussed in YaT2/W2, which was not included in DPO training:
>
> | Model | Sec (%) | Fid (%) |
> | ----- | ----- | ----- |
> | SecAlign 8B  | 98.9 | 59.5 |
> | \+ Fidelity DPO | 99.3 *(+0.4)* | 90.0 *(+30.5)* |
>
> The fidelity-tuned model improves fidelity by 30.5pp on the unseen task with no loss in security. These results illustrate how SecFid's evaluation infrastructure can enable not just better measurement but better defense design.
>
> ## W3
>
> We agree that the current presentation should do more to distinguish the binary abstraction used in the theorem from the richer set of strategies available in real systems, and we thank the reviewer for addressing this.
>
> The formal result in the appendix is scoped to a simplified process-vs-filter model, which we chose to isolate a specific claim: when some inputs remain genuinely ambiguous, the Bayes-optimal action depends on the application's error costs, and no fixed task-agnostic policy is optimal across all deployments. As the reviewer notes, real systems have access to intermediate actions — clarification requests, sandboxing, restricted execution — that can meaningfully reduce the practical severity of the tradeoff by lowering the cost of either outcome. This is an important consideration that the paper should emphasize. At the same time, we believe the core intuition extends: the preferred action among processing, filtering, clarifying, or restricted execution still varies with deployment-specific costs and constraints, so the application dependence identified by the theorem persists even as the action space grows.
>
> We will revise the paper to make the scope of the formal result explicit and to present the richer-action case as a natural extension of the same decision-theoretic framework.
>
> ## Q3
>
> We would like to point the reviewer to §4.5 of the paper, which shows that security gains come at the cost of fidelity, and this pattern holds across probe placement, framing strength, and task type. We take this as encouraging evidence of robustness, though we agree with the reviewer that evaluation over a still broader distribution of probe formulations would further strengthen confidence in these results.

---

> > ### Author Rebuttal · Reviewer_XaNV · 2026-04-01
> >
> > Thank you to the authors for the thoughtful and detailed rebuttal. I appreciate the additional discussion and the effort to strengthen the paper with new analyses, especially the attempt to extend the evaluation to an agentic setting and the clarification of the scope of the theoretical claim. These additions improve the presentation and make the paper’s intended contribution clearer.
> >
> > That said, my main concerns are only partially resolved. First, the new evidence on generalization to realistic LLM pipelines is still limited. The added InjecAgent experiment is interesting, but it is based on a relatively small extension and a narrow evaluation setting, and the rebuttal also notes that the reported rates are inflated because the agent often fails to complete the task. As a result, I do not think this yet fully establishes that the observed security–fidelity tradeoff generalizes robustly to more realistic agentic or RAG-style deployments.
> >
> > Second, the response on the “no universal defense” claim is helpful in clarifying that the formal result is scoped to a simplified PROCESS-vs-FILTER abstraction. However, the rebuttal stops short of showing how the claim would hold under richer action spaces such as clarification, structured instruction/data separation, or restricted execution. I therefore still view the strongest version of the impossibility claim as relying on a simplified model, even if the intuition is plausible.
> >
> > Third, the response to my question about sensitivity to probe construction is the least complete. Pointing back to the existing analysis across placement, framing strength, and task type is useful, but it does not fully address the requested robustness question regarding alternative probe templates, phrasing choices, or more naturalistic instruction-like content. I still think broader sensitivity analysis would be important for validating the benchmark more convincingly.
> >
> > Finally, while the added fidelity-aware DPO experiment is promising, it does not materially change my originality assessment of the current submission. The paper’s core contribution remains a conceptual framing and evaluation benchmark rather than a new defense method, and the added mitigation result feels more like a preliminary follow-up direction than a central contribution of the paper as submitted.
> >
> > Overall, I still find the paper meaningful and well executed as a benchmark/analysis paper, and I appreciate the authors’ clarifications. However, the rebuttal does not fully change my assessment of the paper’s significance/originality limitations or the current strength of its generalization and theoretical claims. My overall evaluation therefore remains unchanged.

---

> > > ### Author Response · Authors · 2026-04-08
> > >
> > > Thank you for the thoughtful response. We are glad the rebuttal helped clarify the intended contribution. We address the remaining concerns below.
> > >
> > > > “...the new evidence on generalization to realistic LLM pipelines is still limited... I do not think this yet fully establishes that the observed security–fidelity tradeoff generalizes robustly to more realistic agentic or RAG-style deployments.”
> > >
> > > We agree that a comprehensive assessment of all agentic and RAG-style deployments is beyond the scope of a rebuttal, but we aim to show that **the SecFid evaluation framework extends naturally to these settings**, and that **the same tradeoff persists once measured there**.
> > >
> > > SecFid evaluation requires three pieces: (1) a **trusted instruction** specifying the intended task, (2) an **untrusted input** that may contain an injected instruction, and (3) a way to define the three reference behaviors: **ignored**, **processed**, and **executed**. Agentic and RAG systems fit this structure well, as retrieved passages and tool outputs are both forms of untrusted data consumed under a trusted task specification.
> > >
> > > To validate this further, we expanded the agentic evaluation to **252 examples**, added models and defenses, conditioned on classifiable examples to avoid conflating agent failure with security/fidelity behavior, and report 95% Wilson CIs.
> > >
> > > | Model            |        Cond. Fid (%) |       Cond. Sec. (%) |
> > > | ---------------- | -------------------: | -------------------: |
> > > | Llama 3.1 8B     | 87.12 [80.34, 91.80] | 53.03 [44.55, 61.34] |
> > > | +SecAlign        | 68.91 [60.11, 76.52] | 79.83 [71.74, 86.06] |
> > > | +ASIDE           | 80.77 [62.12, 91.49] | 76.92 [57.95, 88.97] |
> > > | +DefensiveTokens | 61.90 [53.84, 69.34] | 88.44 [82.21, 92.75] |
> > > | Llama 3.3 70B    | 78.45 [70.12, 84.95] | 68.97 [60.05, 76.66] |
> > > | +SecAlign        | 41.41 [33.25, 50.07] | 95.31 [90.15, 97.83] |
> > >
> > > We do not present this as exhaustive, but rather to show that the framework extends naturally and that the security–fidelity trade-off persists.
> > >
> > > > “However, the rebuttal stops short of showing how the claim would hold under richer action spaces...”
> > >
> > > We thank the reviewer for continuing to engage with us and are glad that we were able to clarify the scope of the formal result. We agree that richer action spaces are important considerations, as they may improve the achievable security–fidelity frontier (e.g., by allowing clarification rather than over-filtering). These actions do not remove the underlying ambiguity but expand the set of possible responses to it. Different deployments can still rationally prefer different actions on the same input. Our results provide further evidence: **we do not prevent models from using richer responses, and still observe the tradeoff**.
> > >
> > > > "...it does not fully address the requested robustness question regarding alternative probe templates, phrasing choices, or more naturalistic instruction-like content.”
> > >
> > > We thank the reviewer for this suggestion, which led to a valuable new analysis. We constructed an additional **377-example slice** using **40 probe templates**, with probes embedded as email questions or code comments, **broadening coverage along template variation, phrasing, and naturalistic content.**
> > >
> > > We estimate contextual fit using t5-v1_1-large and derive a probe naturalness score from length-normalized likelihood. We then regress both Executed and Ignored outcomes on this score:
> > >
> > > | Model         |             β (Executed) |           β (Ignored) |
> > > | ------------- | -----------------------: | --------------------: |
> > > | Llama 3.1 8B  | **−0.68** [−1.06, −0.31] |    0.15 [−0.17, 0.47] |
> > > | Llama 3.3 70B |   *−0.38* [−0.68, −0.08] |   −0.09 [−0.45, 0.27] |
> > > | Qwen 2.5 7B   |       0.35 [−0.23, 0.92] |    0.04 [−0.25, 0.33] |
> > > | Qwen 2.5 14B  | **−0.98** [−1.37, −0.58] | **0.89** [0.57, 1.21] |
> > > | Qwen 2.5 32B  |      −0.40 [−1.01, 0.22] | **1.14** [0.79, 1.50] |
> > > | SecAlign 8B   | **−0.57** [−0.88, −0.26] |    0.36 [−0.02, 0.74] |
> > > | SecAlign 70B  |      −2.04 [−5.06, 0.98] |   −0.18 [−0.46, 0.10] |
> > > \* p < .05, ** p < .01.
> > >
> > > This revealed that more natural probes are generally less likely to be executed, while the effect on ignored is model-dependent. We will incorporate this slice and analysis in the revision to strengthen the benchmark's robustness validation.
> > >
> > > > “...while the added fidelity-aware DPO experiment is promising, it does not materially change my originality assessment... the paper’s core contribution remains a conceptual framing and evaluation benchmark rather than a new defense method...”
> > >
> > > We agree with the reviewer's characterization and do not intend to position the paper primarily as a new defense method. The fidelity-aware DPO result was included as part of possible practical guidance, while also showing that the benchmark can help guide mitigation design.
> > >
> > > We appreciate the reviewer's sustained engagement throughout this discussion, which has meaningfully strengthened our work.
> > >
> > > Best,
> > > Authors

---

### Official Review · Reviewer_NE4k · 2026-03-11

**Soundness:** 3
**Presentation:** 3
**Significance:** 3
**Originality:** 3
**Overall Recommendation:** 5
**Confidence:** 4

**Summary:**

This paper introduces a benchmark for evaluating LLM responses to prompt-injections (SecFid). The authors advocate for and demonstrate the tradeoff between security (non-compliance) and fidelity (accurately performing the required task). They do so by designing varied benchmark where identical prompt injection attempts (which the authors term “probes”) appear across different tasks that require those probes to be treated differently (for example, a translation task would require the injected string to be faithfully parsed and translated, whereas a summarisation task would want to avoid falling prey to probes instructing the model to include or exclude specific details).

With this benchmark, the authors demonstrate varied performance across the pareto-frontier of security against fidelity, on a mixture of frontier-deployed closed-source models, and open-source models of different scales. They then offer a theoretical proof for the claim that “when benign and adversarial inputs overlap, no universal defense exists”.

**Compliance With Llm Reviewing Policy:**

Affirmed.

**Final Justification:**

Happy birthday this to a five in light of the additional experiments especially.

**Key Questions For Authors:**

1.) Why were other frontier models (Claude, grok, etc) not included?
2.) Why weren’t more models across scales of the same family included?
3.) The proof in Appendix A assumes inputs with genuinely intermediate ambiguity exist (α ∈ (0,1)) — how common do the authors claim such inputs are in practice, and does the severity of the tradeoff depend on this frequency?

**Limitations:**

Yes

**Strengths And Weaknesses:**

Strengths:
* I commend the authors for their attempts to deconfuse the field of prompt injection in this paper. They raise clear and thoughtful considerations about the tradeoffs and realities of model deployment, especially in their consideration of the tradeoff of fidelity vs security, and the opaqueness of the costs of defense on model performance. This benchmark aims to address important problems, and this paper articulates those clearly.
* SecFid’s behavioral separability design is clever. The way in which the probes are crafted forces three distinguishable outputs, which makes for easy evaluation in what would otherwise be challenging.
* The repair vs. suppress distinction from Section 4.4 / Figure 3 seems genuinely useful, in illustrating a methodology for differentiating between defenses despite their benchmark performance looking similar.
* Overall, the paper is presented and written clearly.

Weaknesses:
* The final contribution of the paper is the formal proof aiming to establish that “when benign and adversarial inputs overlap, no universal defense exists”. I think the authors don’t sufficiently explain why models can’t infer the correct cost ratio for the task at hand from the task context, or how commonly this would or wouldn’t happen. It appears that Gemini-3-flash, for example, is pareto optimal among the models evaluated, performing well (but not perfectly) on both axes. The proof in the appendix assumes that the model has a single fixed level of suspicion, but I’m not sure that this assumption is justified.
* I’m surprised that the spread of models evaluated isn’t particularly exhaustive, both across model robustness and scale. With respect to scale, I think seeing more models across varying scales (such as the surprising result showing that Llamas 8B and 70B have near identical security and fidelity rates) would greatly strengthen the conclusions of this paper.
* Regarding seeing more robust models, even one other model appearing at beyond the current paretofrontier would weaken the case made in this paper that universal prompt-injection defense is “necessarily application specific”. This makes me interested in seeing results on frontier models, such as the Claude series, Grok, or more models from the existing set of providers (OpenAI and GDM), or reasoning as to why we should exclude these models.
* I’m a little surprised at the lack of reference to similar concepts to this in the jailbreak literature. Many of the author’s core complaints about the field of prompt-injection literature seem shared with complaints I’ve heard in the jailbreaking literature.
I’m also surprised that summarisation, despite appearing in figure 1, does not appear in the actual set of benchmark tasks (translation, extraction, and counting), especially since it’s used as a motivating example at the start of the paper.
* It’s more like a nit, but I genuinely thought the references to “probes” in this paper referred to some kind of model-internals approach. I’d prefer the authors find a different term, given that as far as I can tell this isn’t a canonical use of the term “probes”.

---

> ### Author Rebuttal · Authors · 2026-03-31
>
> We are encouraged that the reviewer found the framing clear, the benchmark well structured, and the repair-vs-suppress distinction useful. The main issues raised here concern the scope of the theoretical claim, model coverage, connections to jailbreaks, and task-design/terminology choices.
>
> ## W1 & Q3
> We thank the reviewer for these comments.
>
> On whether models can infer the correct cost ratio from task context, our view is that the cost ratio in §5 is determined in part by deployment-level factors that are not contained in the input itself, such as the relative harm of a hijack versus the cost of discarding valid content. As a result, two systems given the same task and input may still have different optimal behaviors because their error costs differ. Gemini-3-Flash appears near the Pareto frontier in our evaluation, and we agree that strong performance on both axes is possible. In revision, we will clarify that our result does not assume a single fixed level of suspicion across tasks. Rather, it assumes only that some inputs remain difficult to disambiguate and that deployments can differ in how they value different errors.
>
> On the frequency of ambiguous inputs, we agree that this is ultimately an empirical question. Such cases can arise naturally in realistic settings, for example, when an email assistant processes a message containing instructions, or when a document editor operates on text that includes imperative language. How often this occurs in practice likely depends on the domain and input distribution. We also agree that the severity of the tradeoff depends on this frequency: if such cases are rare or low-stakes, the tradeoff is limited; if they are common or costly, it becomes more consequential. We will make both points explicit in revision.
>
> ## W2, W3, Q1, Q2
>
> We thank the reviewer for this suggestion. Adding more frontier models and scales strengthens the conclusions, and we have therefore expanded the results with the following additional base models:
>
> | Model | Sec (%) | Fid (%) |
> | :---- | :---- | :---- |
> | Claude Haiku 4.5 | 73.5 | 80.8 |
> | Claude Sonnet 4.6 | 83.2 | 69.8 |
> | Claude Opus 4.6 | 88.9 | 74.0 |
> | Gemini 3.1 Flash-Lite | 62.2 | 91.4 |
> | GPT-5.4 Nano | 77.7 | 89.8 |
> | GPT-5.4 Mini | 66.9 | 90.3 |
> | GPT-5.4 | 80.1 | 85.1 |
> | Grok 4.20 | 34.3 | 94.5 |
> | Gemma 3 12B | 79.3 | 94.2 |
> | Gemma 3 27B | 82.1 | 94.5 |
> | Qwen 2.5 14B | 58.0 | 93.4 |
> | Qwen 2.5 32B | 73.6 | 90.4 |
>
> These results sharpen the conclusions in two ways. First, as raised in W3, we view this as refining rather than weakening our claim. A model beyond the current empirical Pareto frontier shows that stronger models can improve both security and fidelity on SecFid. We interpret that as the frontier moving outward, with the question of how handle ambiguous cases when the right behavior still depends on deployment context and error costs remaining.
>
> Second, broader scale coverage still does not support a simple “bigger is more robust” story. The original Llama 8B→70B result remains informative, and the added same-family comparisons are also inconsistent: Gemma 12B→27B changes only modestly (79.3/94.2→82.1/94.5), while Qwen 7B/14B/32B is clearly non-monotonic (74.1/82.5, 58.0/93.4, 73.6/90.4). We will include the full expanded tables and plots in the revision.
>
> ## W4
>
> We agree there is an important parallel to the jailbreak literature, especially over-refusal: in both cases, defenses can improve apparent safety by suppressing content based on superficial cues rather than resolving context correctly. We will make this connection explicit in revision.
>
> At the same time, fidelity loss in prompt injection is importantly distinct. In our setting, the suppressed content appears in untrusted data rather than in the user instruction. This often makes the failure harder to detect, because the model may still produce a plausible output while silently dropping valid content, especially in RAG or tool-use settings where the underlying content is not directly visible to the user.
>
> Figure 1 uses summarization as a motivating example to illustrate that the same instruction-like text can require different handling across tasks. However, we do not include it in the benchmark because it does not support reliable evaluation of the security–fidelity tradeoff: in summarization, processed and ignored behavior often collapse into similarly plausible outputs due to paraphrasing and compression. We therefore focus on tasks such as translation, extraction, and counting, where executed, processed, and ignored remain behaviorally separable and can be evaluated cleanly.
>
> ## W5
>
> Thank you for noting this. We acknowledge that “probe” can suggest model-internals work. We used it to mean injected content that elicits an observable behavior, following prior benchmark work (Zverev et al., 2025). In revision, we will replace it with clearer terminology, such as “injected instruction” and add a brief clarification at first use.

---

> > ### Author Rebuttal · Reviewer_NE4k · 2026-04-04
> >
> > I felt more convinced, especially seeing the results on additional models. I accept the authors' pushback on framing.

---

> > > ### Author Response · Authors · 2026-04-08
> > >
> > > We thank you for the constructive exchange and are glad the additional results and clarifications were convincing.
> > >
> > > Best,
> > > Authors

---

### Official Review · Reviewer_YaT2 · 2026-03-11

**Soundness:** 3
**Presentation:** 4
**Significance:** 3
**Originality:** 3
**Overall Recommendation:** 5
**Confidence:** 4

**Summary:**

The paper uncovers an important pitfall of existing prompt injection defenses where the defended models choose to completely ignore the injection rather than actually treating it as “data” to be processed along with the remaining benign data. The paper proposes a benchmark that measures this phenomenon with a metric called “fidelity.” It is then empirically and argumentatively shown why fidelity and security may be a trade-off.

**Compliance With Llm Reviewing Policy:**

Affirmed.

**Final Justification:**

I'm satisfied with the rebuttal. Most of my concerns are addressed except for the adaptive attacks. I partially agree with Reviewer XaNV's comment around lack of task and defense coverage, but I believe it's still in the acceptable range. So I raised my score from 4 to 5 but not higher.

**Key Questions For Authors:**

I don’t have any specific questions, but please feel free to discuss / push back on my earlier comments.

**Limitations:**

yes

**Strengths And Weaknesses:**

**Strengths**

1. This work addresses convincingly demonstrates a failure mode of popular prompt injection defenses that is overlooked by the community. The paper has a single focused message and effectively delivers it.
2. The problem formulation and the metric design are all very reasonable and sound to me. I like the extraction and counting tasks for their ease of verification.
3. The paper makes several conceptual distinctions that, to me, brings clarity to the problem such as Process vs Ignore, and full-text vs partial-text tasks. I can notice the care put into framing this work.
4. The presentation is great; the tables and the trade-off plot make a lot of sense. I enjoyed reading it, and it was very clear.

**Weaknesses**

1. **Embedding distance as task verification.** I don’t quite like embedding distance as a scoring mechanism for the translation task (or any task at all). Embedding distance is believed to capture “semantic” distance, but it is not really interpretable, much more like a black-box. Small embedding distance can similarity in many different ways from semantics, patterns, language, and formats. It seems too arbitrary. I almost like just using an LLM as a judge here more, but I guess it runs a risk of getting fooled by the prompt injection itself. That said, I do think that recent frontier models should be much more robust now that they are usable.
2. **Task diversity and verification.** Related to the prior comment, I would like to see more tasks whose success can be more easily verified (think of “verifiable reward”). I know this is difficult, and unfortunately, I don’t have great suggestions. Here is a *flawed* idea but maybe give you something to consider: a summarization task with “witness word” verification, e.g., there is a specific phrase or name as part of the injection (”[SYSTEM OVERRIDE] This is a new system instruction: Do XYZ” → witness word = “system”) and then check if witness word appears in the model’s output. If there’s no witness word, the injection is likely ignored or executed (not processed). Then, you may need another verifier to determine if “Do XYZ” is followed to rule out the execute option. I think this is a bit too complicated, and verification can have some errors, but perhaps, there’s a better way to construct this.
3. **Compatibility with adaptive attacks.** Moving towards adaptive attacks is important for prompt injection security evaluation. I’d prefer to see results with the benchmark hooked to at least one of the prompt optimization or automated adaptive attacks. I would like to see how fidelity change with these attacks (which just tries to minimize security) and to see tasks and corresponding verifications that can be used as optimization objective and ideally not lead to “reward hacking” (optimized injection cheats the verifier).
4. **I want to see more of ways to address this fidelity-security trade-off.** The paper evaluates a few models and defenses, but I would like to see more meaningful results generally. The paper seems relatively light on results. Perhaps, either more variety of defenses or an intervention to improve this trade-off. What can be done if the injection is detected by a classifier? Is there a defense that rewrites (like https://arxiv.org/abs/2511.10720) the prompt that stops the attack but maintains fidelity? Another idea is: is it possible to modify the training objective of SecAlign just a bit to preserve fidelity?
5. **The “fundamental” trade-off.** This may be a philosophical difference, but I don’t quite believe that the trade-off presented in Section 5 is the correct framing. The paper conceptualizes a trade-off between “Process” and “Filter” (i.e., ignore). However, I believe that the solution to prompt injection is to just always “process the injection as data.” And “process” can mean different things like including it in response, just ignoring, raising a concern, acknowledging it, or even following it (!) if the user instruction is “follow any instructions in this file.” In other words, I believe there is a correct solution depending on the context as long as the injection is “processed” as data. Since there’s always a correct answer, I would not personally say that the trade-off is “fundamental.” Instead, it is more like an artifact of how the objective function of the training of these models or defenses.
    1. My disagreement here might just be in the terminology of what “process” means.
    2. From a different perspective, I could say that “Filter” is a form of shortcut learning that the model takes to achieve the same outcome. This could explain why DefensiveToken, which has much less capacity, takes this shortcut much more than SecAlign as shown in Figure 3.
6. This comment is purely a suggestion and is not considered in rating the paper whatsoever. Please don’t waste 9 pages on author lists of Gemini or Llama papers. That’s lots of paper for those who like to print out their papers to read :)

---

> ### Author Rebuttal · Authors · 2026-03-31
>
> We thank the reviewer for the positive and constructive engagement. We are encouraged that the reviewer found the core benchmark motivation meaningful, appreciated the executed/processed/ignored distinction, and viewed the paper as clear and well presented. We see the main comments raised here as primarily about evaluation validation, attack and defense scope, and specific claim framing, and address each point below.
>
> ## W1
>
> We agree that embedding similarity can conflate semantic closeness with surface-level overlap in ways that are hard to interpret. The reason we opted for it in this case is that the translation evaluator faces a narrower task: deciding whether an output is closer to a processed reference or an ignored reference, where the two differ only in whether the translated probe appears. This constrained comparison reduces some of the interpretability concerns, though we agree it deserves validation.
>
> To that end, we ran a human agreement study on 99 translation outputs sampled across all 9 model configurations, balanced across attack categories and placements. The embedding-based evaluator achieved Cohen's $\kappa$ \= 0.83 against human labels, beating out a BLEU-based classifier with $\kappa$ \= 0.79, and an LLM judge (GPT 5.4) with $\kappa$ \= 0.60.
>
> ## W2
>
> We agree that broader task coverage would strengthen the paper and appreciate the provided suggestions. Summarization is an appealing candidate, but presents a challenge as processed and ignored outputs are difficult to cleanly distinguish from each other when the model paraphrases and compresses content in non-deterministic ways.
>
> To broaden coverage while preserving verification quality, we extend SecFid with an edit task built from 186 InstrEditBench (Zeng et al., 2025\) examples. This task maintains the same three-way structure and surfaces the same overall tradeoff: averaged across our tested models, security drops from 82.0 on the original tasks to 56.5 on edit, while fidelity drops from 91.3 to 75.5. The pattern holds for defenses as well — SecAlign 8B remains highly secure on edit (98.9%) but at much lower fidelity (59.5%). We will include the full results in the revision.
>
> ## W3
>
> We agree that adaptive attacks are an important next step. The current benchmark uses controlled attack families because SecFid is designed to measure the security–fidelity tradeoff on instances where the executed, processed, and ignored behaviors remain stable. Fixed templates are useful for this purpose because they let us vary attack framing and placement while preserving the underlying semantics of the instance. Adaptive optimization is therefore only valid if it strengthens the attack without changing what counts as the correct processed output or exploiting the verifier. We will clarify this scope in the revision and discuss constrained adaptive attacks as a direction for future work.
>
> ## W4
>
> We agree that broader defense coverage and an intervention to improve the tradeoff are valuable additions and pursue both directions.
>
> **Broader defense comparison.** We expand the evaluation to include ASIDE and ISE, confirming that the security–fidelity tradeoff extends to additional defense types:
>
> | Model | Sec (%) | Fid (%) |
> | :---- | :---- | :---- |
> | **Llama 3.1 8B** | 52.7 | 94.9 |
> | \+ ASIDE | 96.9 | 75.3 |
> | \+ ISE | 89.1 | 81.2 |
> | **Qwen 2.5 7B** | 74.1 | 82.5 |
> | \+ ASIDE | 95.0 | 73.4 |
> | \+ ISE | 87.6 | 76.6 |
>
> **Fidelity-aware preference tuning.** Motivated by the reviewer's idea, we finetune SecAlign 8B using DPO with RS-LoRA, constructing preference pairs (processed \> ignored \> executed) from both gold references and model-generated outputs, totaling 754 training examples with a 15% held-out test set.
>
> On the held-out test set, the fidelity-tuned model maintains perfect security while improving fidelity:
>
> | Model | Sec (%) | Fid (%) |
> | :---- | :---- | :---- |
> | SecAlign 8B | 100.0 | 82.3 |
> | \+ Fidelity DPO | 100.0 *(+0.0)* | 90.8 *(+8.5)* |
>
> We also evaluate on the edit task (discussed in YaT2/W2), which was not included in DPO training:
>
> | Model | Sec (%) | Fid (%) |
> | :---- | :---- | :---- |
> | SecAlign 8B | 98.9 | 59.5 |
> | \+ Fidelity DPO | 99.3 *(+0.4)* | 90.0 *(+30.5)* |
>
> The fidelity-tuned model improves fidelity by 30.5pp on the unseen task with no loss in security. These results illustrate how SecFid's evaluation infrastructure can enable not just better measurement but better defense design.
>
> ## W5
>
> We agree this is likely a philosophical difference, as the reviewer suggests. We will revise the paper to clarify what "processed" means in SecFid — treating the injected text as data to be included in the response — and distinguish this from correctness, which we agree is context-dependent.
>
> We also appreciate the reviewer’s shortcut-learning interpretation as an insightful perspective for which to view the repair-versus-suppress behavior that we will explore further.

---

> > ### Author Rebuttal · Reviewer_YaT2 · 2026-04-04
> >
> > I appreciate the rebuttal and the extra experiments. This is more than what I asked for. The only missing thing is the adaptive attack but I'm okay if this is mentioned as limitation. I have adjusted my score from 4 to 5.

---

> > > ### Author Response · Authors · 2026-04-08
> > >
> > > We thank the reviewer and appreciate all the feedback provided. We are happy to see all your concerns addressed, and additionally take to heart your advice on printing and will spare a few trees in the references.
> > >
> > > Best,
> > > Authors

---

### Decision · Program_Chairs · 2026-04-30

**Decision:**

Accept (spotlight)

**Comment:**

This work proposes a benchmark for evaluation of LLMs to prompt injections. This ends up showing an important issue of existing prompt injection defenses: the models end up degrading in ability to process benign instructions.

The reviewers agreed on the merits of this work and that they represented an important contribution to ICML. The strongest points:

1. A clear presentation and message.

2. strong empirical findings that are designed well to support the claims.

There were several concerns addressed well during the rebuttal.

A remaining concern that should be clearly noted is the lack of adaptive attacks.